# Artificial Intelligence Techniques and Pedigree Charts in Oncogenetics: Towards an Experimental Multioutput Software System for Digitization and Risk Prediction

**Luana Conte** [1,2], **Emanuele Rizzo** [3,*], **Tiziana Grassi** [4], **Francesco Bagordo** [5], **Elisabetta De Matteis** [6] and **Giorgio De Nunzio** [1,2]

1. Laboratory of Biomedical Physics and Environment, Department of Mathematics and Physics "E. De Giorgi", University of Salento, 73100 Lecce, Italy
2. Laboratory of Advanced Data Analysis for Medicine (ADAM), Laboratory of Interdisciplinary Research Applied to Medicine, University of Salento/Local Health Authority of Lecce, 73100 Lecce, Italy
3. Department of Biological and Environmental Sciences and Technologies, University of Salento, 73100 Lecce, Italy
4. Department of Experimental Medicine, University of Salento, 73100 Lecce, Italy
5. Department of Pharmacy-Pharmaceutical Sciences, University of Bari "Aldo Moro", 70124 Bari, Italy
6. Oncological Screenings Unit, Local Health Authority of Lecce, 73100 Lecce, Italy
* Correspondence: emanuele.rizzo1@unisalento.it

**Abstract:** Pedigree charts remain essential in oncological genetic counseling for identifying individuals with an increased risk of developing hereditary tumors. However, this valuable data source often remains confined to paper files, going unused. We propose a computer-aided detection/diagnosis system, based on machine learning and deep learning techniques, capable of the following: (1) assisting genetic oncologists in digitizing paper-based pedigree charts, and in generating new digital ones, and (2) automatically predicting the genetic predisposition risk directly from these digital pedigree charts. To the best of our knowledge, there are no similar studies in the current literature, and consequently, no utilization of software based on artificial intelligence on pedigree charts has been made public yet. By incorporating medical images and other data from omics sciences, there is also a fertile ground for training additional artificial intelligence systems, broadening the software predictive capabilities. We plan to bridge the gap between scientific advancements and practical implementation by modernizing and enhancing existing oncological genetic counseling services. This would mark the pioneering development of an AI-based application designed to enhance various aspects of genetic counseling, leading to improved patient care and advancements in the field of oncogenetics.

**Keywords:** artificial intelligence; machine learning; deep learning; pedigree charts; oncogenetics; oncological genetic counseling

## 1. Introduction

Despite the continuous scientific progress that has occurred in recent decades, cancer is still a leading cause of death and a public health concern worldwide. The latest available data (2020) report an estimated 18.1 million new cancer cases (excluding non-melanoma skin cancer) [1], with nearly 10 million deaths [2].

Cancer is a large set of diseases that have in common an altered cellular genomics, in a multifactorial etiology context. Depending on their onset, tumors can be classified in sporadic, familial, and inherited ones, the latter accounting for about 10% of all diagnoses [3]. Hereditary cancers are mainly due to pathogenic autosomal dominant mutations in high-penetrance susceptibility genes; there are known cancer driver genes that function as tumor suppressors, oncogenes, or have a role in maintaining DNA stability, usually associated with DNA repair and homologous recombination pathways [4]. Examples are genes such as those responsible for Hereditary Breast and Ovarian Cancer (HBOC) (BRCA1 and

BRCA2) and Lynch syndrome (MLH1, MSH2, MSH6, and PMS2), whose discovery dates back to the early 1990s [5]. Since then, thanks to the advent of Next Generation Sequencing (NGS) and Genome-Wide Association Studies (GWAS), other Cancer Predisposition Genes (CPGs) have been cloned and isolated, both with high and medium–low penetrance; many of these have entered clinical practice in the so-called multigenic panels, which today tend to be preferred to single genetic analyses.

Oncological clinical genetics, also known as oncogenetics, is therefore a rather young science and with it, the clinical processes that lead to the diagnosis of hereditary cancer and the management of subjects at high genetic risk of cancer (both at a preventive and therapeutic level) have been developed. Nowadays, these principles have merged into the so-called Oncological Genetic Counseling (OGC), a well-established practice within all oncology services [5]. The first act of the traditional OGC (pre-test session) consists of the reconstruction of the personal and family history. Furthermore, in addition to the acquisition of the clinical documentation of the reported cancer cases, the complete pedigree chart is reconstructed, at least up to the third degree of kinship [5]. At this point, after the choice of the most suitable subject for the test (named "index case" or "proband"), a single or multigene germinal analysis is carried out, normally on peripheral blood, and the result is awaited, to be communicated together with the possible surveillance plan (post-test session). In other words, from the pre-test session onwards, genetic analyses are the focus of the clinician's attention, with the result that pedigree charts are rarely resumed and updated and, often in a paper format, remain in archives with a limited use. But these diagrams are full of useful information; they usually report all tumor diseases (sometimes also others of non-neoplastic nature) along with the age of onset, the results of any genetic test previously performed on the subjects included in the various generations, the age of eventual death, etc. Hence, there is a need to recover and rework these pieces of information and make them available to health professionals and researchers, possibly with an innovative approach.

Artificial Intelligence (AI) techniques and technologies are probably the best tool from this point of view. AI could push towards a direct digitization of these data, today usually available in a semi-digital format [6]. Moreover, AI, by its analytical power and ability to deduce elements of order in complex systems, can provide a decisive contribution to a better understanding of the pathogenesis of hereditary and sporadic tumors, as well as of any association between genes, mutations, tumors, other diseases, and personal characteristics. Furthermore, digitized pedigree charts can also be utilized as a training dataset for AI models in order to perform correlation studies and predict the risk of contracting a neoplastic disease or infer the presence of a pathogenic mutation in a specific gene, as statistical software such as IBIS (International Breast Cancer Intervention Study-Online Tyrer-Cuzick model breast cancer risk evaluation tool), BRCAPRO and BOADICEA (Breast and Ovarian Analysis of Disease Incidence and Carrier Estimation Algorithm) currently perform based on some personal parameters [7]. Several studies in oncology have been carried out by AI, ranging from images, genomics, personalized medicine, multi-omics, and drug discovery [8–10], as well as diagnostic algorithms like those developed by Google DeepMind or intelligent assistants such as IBM's Watson for Oncology [11,12].

CAD (Computer-Aided Detection/Diagnosis) systems are digital instruments which have emerged over the last 20 years to cope with problems related to the limits of human perception [13], aiming to analyze and process various types of medical images and produce an automatic or semi-automatic diagnosis [14]. These tools are often combined with AI (in particular with Machine Learning (ML)) in order to achieve the most reliable output possible [15]. CAD systems are increasingly used in the medical practice, above all, in oncology; this is demonstrated by the thousands of papers published in recent years, of which, over a hundred were published in 2023 alone (Table 1).

**Table 1.** Most influential papers on CAD systems in oncology published in 2023.

| Study | Study Type | CAD Application |
| --- | --- | --- |
| Kumar, S. et al. [16] | Review | Breast cancer diagnosis |
| Singh, S. et al. [17] | Review | Hepatocellular carcinoma diagnosis |
| Maida, M. et al. [18] | Review | Endoscopic screening of colorectal cancer |
| Loizidou, K. et al. [19] | Review | Breast cancer detection and classification in mammography |
| Ramaekers, M. et al. [20] | Review | Pancreatic cancer diagnosis |
| Ranjbarzadeh, R. et al. [21] | Review | Breast tumor localization and segmentation |
| Jenkin Suji, R. et al. [22] | Review | Lung segmentation and nodule detection in CT images |
| Raghavendra, U. et al. [23] | Review | Brain tumor detection and screening |
| Hu, L. et al. [24] | Research article | Assessment of MRI-visible prostate cancer |
| Kim, H. et al. [25] | Research article | Screening outcomes of digital mammography |
| Mansur, A. et al. [26] | Review | Risk prediction and therapy response in colorectal cancer |
| Nicosia, L. et al. [27] | Research article | Assessment of breast ultrasound lesions |
| Ali, Z. et al. [28] | Research article | Melanoma lesions segmentation |

Nonetheless, to the best of our knowledge, no research article concerning the digitization of pedigree charts exists in the scientific literature to date and no software tools capable of predicting genetic predisposition risk directly from digital pedigree charts, or derived from a paper-based source, have yet to be made available. A system will thus be conceived from scratch, following state-of-the-art techniques from the domain of computer vision, and taking inspiration from similar problems already known and reported.

A major limiting factor in this issue is symbol recognition in hand-drawn graphs. A review of the relevant literature led to discover various related fields of interest which can give suggestions on potentially successful approaches. They cover the problems of automatic hand-drawn graphical schemes and diagram digitization, from the recognition of the symbols adopted in the graphs, to the reconstruction of the underlying logical structure. Some computer vision and understanding problems that we considered similar to the pedigree chart digitization one are generic symbol recognition, UML (Unified Modeling Language) graph digitization, flowchart plots, logic circuit analysis, piping and instrumentation diagrams, electrical circuits, generic engineering drawings, and chemical structure recognition. All these problems share the need to convert hand-drawn graphs, composed of symbols with (inside or outside) text annotations connected by (single or multiple) edges, to a digital format.

Among the most recent and interesting papers concerning generic symbol recognition in hand-drawn diagrams/graphs, it is important to cite [29], in which the YOLSO (You Only Look for a Symbol Once) engine is described, a specialized single-stage object detector for fixed-size, non-uniform symbols in maps and historical documents, using a unique grid approach and fully convolutional architecture for map digitization. It also offers coarse segmentation. It is a specialized form of YOLO (You Only Look Once) [30], used in many fields of computer vision, including the recognition of handwritten diagrams: for example, in [31], the authors focus on recognizing handwritten diagrams, particularly finite automata images, exploring the use of YOLO and YOLO-Tiny networks for symbol detection and bit-string processing, and achieving promising results with 82.04% average precision and 97.20% recall in detecting finite automata symbols in a handwritten dataset.

Concerning symbol recognition and graph reconstruction for UML diagrams, [32] discusses the need for a software that can automatically recognize and convert hand-drawn engineering diagrams into a format compatible with CASE (Computer-Aided Software Engineering) tools, reducing manual efforts and time in identifying vulnerabilities and errors. The authors then propose a solution based on image processing algorithms such as corner detection, link detection, and classification by descriptors.

Piping and Instrumentation Diagrams (P&IDs) are another interesting field for automatic digitization. For example, recent works [33,34] present methods respectively using an improved continuous line detection algorithm to extract information from P&IDs, in-

cluding identifying connection relationships and creating digital P&IDs, and an end-to-end digitization method based on Deep Neural Networks (DNNs) for converting P&IDs into digital form by object recognition, topology reconstruction, and diagram generation. A review of papers on hand-drawn chemical structure reconstruction can be found in [35].

There are two recent works concerning the reconstruction of electrical and logical circuits [36,37]. The former discusses new advancements in using neural networks for automatically generating simulation-ready electronic circuits from hand-drawn circuit diagrams; the proposed algorithm first identifies circuit components using YOLOv5 object detection, achieving a high accuracy rate of 98.2%. Subsequently, it reconstructs the circuit schematic using a novel Hough transform-based approach for node recognition. On the other hand, [37] addresses the relatively understudied area of analyzing handwritten logic circuits; again, a DNN based on YOLO is used to identify the circuit components within the handwritten diagram, then a simple boundary tracking method is employed to recognize the connections among these identified components. The results indicate that the YOLO algorithm outperformed other Deep Learning (DL) methods such as Faster R-CNN, Detectron2, and RetinaNet in identifying logic gates within the proposed system.

The field of automatic flowchart digitization is quite rich. Among recent papers, [38] focuses on recovering handwritten flowcharts used in algorithm programming and design and outlines a pipeline for recognizing elements within handwritten flowcharts through Convolutional Neural Networks (CNNs), generating a digitized version of the flow diagram and the C programming code that corresponds to the recognized flowchart. In [39], the authors introduce an end-to-end multi-task network called FR-DETR (Flowchart Recognition DEtection TRansformer) and present a new dataset designed to enhance the precision and robustness of flowchart recognition. FR-DETR consists of a Convolutional Neural Network backbone and a shared multi-scale transformer structure. It performs both symbol and edge detection by utilizing shared feature maps and separate prediction heads. The recognition process follows a coarse-to-fine refinement approach. The experimental results demonstrate the effectiveness of FR-DETR. It achieves an overall precision of 94.0% and a recall of 93.1% on the newly proposed dataset and even higher precision (98.7%) and recall (98.1%) on the CLEF-IP (Conference and Labs of the Evaluation Forum—Intellectual Property) dataset, outperforming previous methods in flowchart recognition. Finally, in [40], the proposed process involves several steps, in particular, shape extraction achieved using the Otsu thresholding algorithm, various morphological operations, including erosion, dilation, and opening, applied to refine the extracted shapes and enhance their quality and accuracy, polygon approximation (smoothening the shape outlines) by the Ramer–Douglas–Peucker algorithm, and handwriting recognition to identify and process the text inside the blocks of the flowchart. Some of the papers, in particular [38,40], provide their software, which can be used as a starting point in our experiments.

This article is intended as a research plan and is subdivided as follows: in the Materials and Methods section, the planned characteristics of our CAD tool will be defined, with details of our approach for the digitization of hand-drawn pedigree charts and some simple feasibility tests, alongside the automatic prediction of individuals at risk of genetic mutation directly on the digitized pedigree charts. Later, the expected results of our CAD will be described in the Results section and discussed in the Discussion section.

## 2. Materials and Methods

### 2.1. Hand-Drawn Pedigree Chart Digitization: Our Approach

Figure 1 shows a sample of a pedigree chart from our database. The difficulties of the automatic digitization task are evident: scan quality (e.g., document skew/rotation, uneven illumination, noisy background), variability caused by individual writing style, complex layout, inconsistent spacing in text, poor text legibility, and deformation in symbol shapes. All these issues and others suggest that the digitization process should be necessarily interactive, asking for the realization of a system for rough chart symbol, relationships, and text recognition, followed by manual refinement.

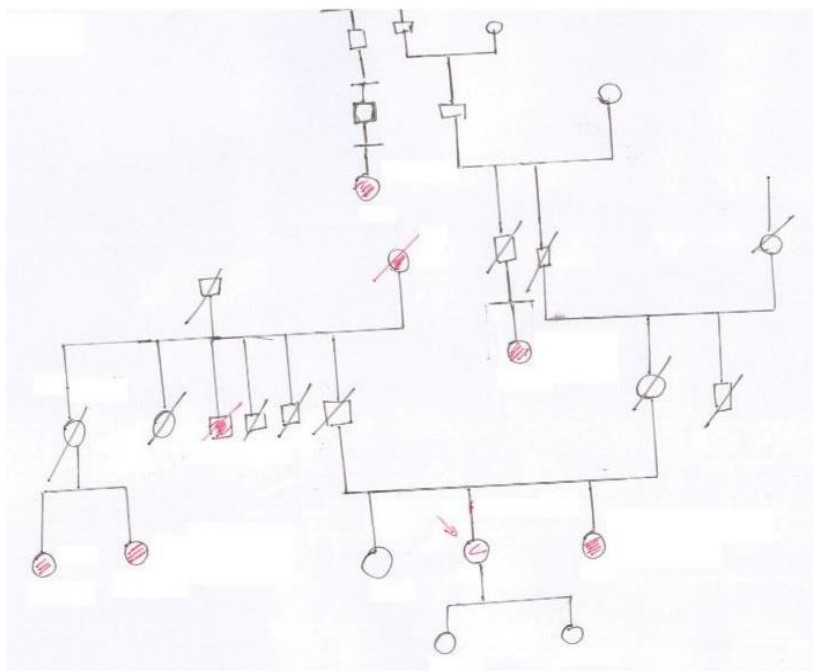

**Figure 1.** A sample of a pedigree chart hand-drawn on paper. The present graph may not reflect the official symbology.

Figure 2, obtained from https://opengenetics.pressbooks.tru.ca/chapter/pedigree-analysis/#LongDesc4.2.1Symbols (accessed on 26 September 2023), shows some symbols used in pedigree charts.

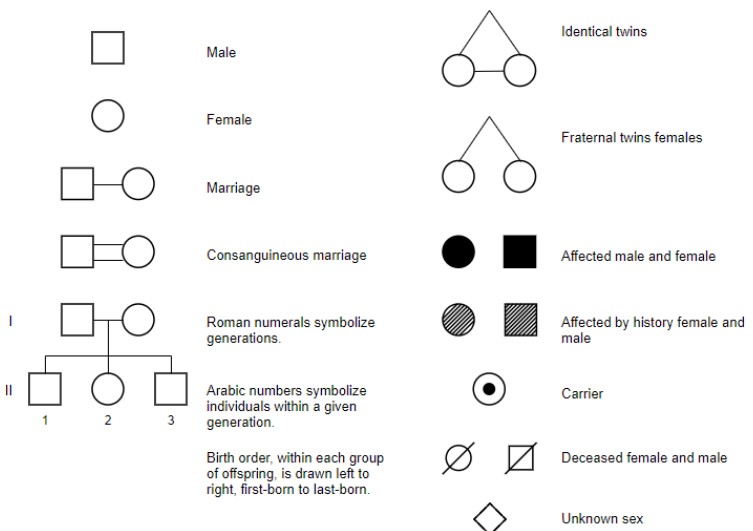

**Figure 2.** Some standard symbols used in constructing a pedigree chart.

Schematically, and disregarding old-fashioned procedures based on the recognition of symbols by explicit feature calculations (as these methods are strongly dependent on the symbol shape cleanness and tend to lack sufficient robustness), the approach we will follow consists of a typical supervised scheme based on DL object detection and recognition, in which each pedigree symbol would be treated as a different class label for classification, and the model would identify which symbols are present and their spatial relationships to reconstruct the pedigree chart structure.

As anticipated, it will be engineered as a semi-interactive tool so to allow error correction. A preliminary realization step will be the creation of a training database of hand-drawn pedigree chart symbols compliant to the standard human pedigree nomenclature recommended by the Pedigree Standardization Work Group or PSWG (formerly Pedigree Standardization Task Force, PSTF) of the National Society of Genetic Counselors (NSGC) [41–45]. In this standard, adopted worldwide, nomenclature and recommended pedigree symbols, to be used in clinical practice, publications, and electronic health records, are established. The most recent standard can be found in [43] and is based substantially on the use of simple shapes (squares for men, circles for women, 45-degree rotated squares for non-binary) to denote family member sex, with text annotations to specify age, gender, dates of birth/death, cause of death, etc., and variations such as filled shapes to indicate clinically affected individuals, and diagonal lines through the symbols to specify diseased individuals. Beside symbols and text, pedigree charts also rely on connecting lines to specify relationships and thus the flow of genetic information within a family is as follows: horizontal lines are used to represent marriages or partnerships between two individuals, vertical lines descend from the marital union line to connect parents with their children, siblings are connected by horizontal lines that run parallel to each other and connect to their parents' vertical lines, etc.

## 2.2. Procedure for Automatic Pedigree Chart Reconstruction

The procedure for automatic pedigree reconstruction for paper charts will consist of the following steps:

- Pedigree chart scanning, producing image files from paper charts (manual step).
- Automatic recognition of pedigree symbols, line segments, and text parts.
- First correction module: interactive step to improve recognition by adding/refining/removing symbols, line segments, text, which were not correctly recognized (manual step).
- Connectivity recognition and structure analysis, i.e., finding connections between symbols, line segments, and text parts, to model the pedigree chart.
- Second correction module: manual intervention to refine the reconstructed chart;
- Saving to a standard format for subsequent elaboration.

The procedure will be mostly implemented in the Python language, possibly with other convenient languages/environments for the Graphical User Interface (GUI).

Some steps of the reported procedure deserve more details.

## 2.3. Automatic Recognition of Symbols, Edges, and Text

Many state-of-the-art software tools exist for Optical Character Recognition (OCR), so a choice of already working solutions will be tested. Among them, Tesseract OCR (https://github.com/tesseract-ocr/tesseract) (accessed on 27 September 2023) is one of the most widely used open-source OCR engines; it is developed by Google and provides support for multiple languages; its direct Python wrapper is the pytesseract library. Tesseract can be found, together with other utilities for OCR pre-processing and post-processing, in OCRopus, which is a collection of OCR-related tools (https://github.com/tesseract-ocr/ocropy) (accessed on 27 September 2023). Another Python wrapper for tesseract and other OCR engines, such as CuneiForm and GOCR, is PyOCR (https://github.com/jflesch/pyocr) (accessed on 27 September 2023), which allows the use of multiple OCR engines from a single interface. Finally, EasyOCR is a DL-based library for Python, supporting multiple languages and known for its ease of use. The reported list is not exhaustive but gives an idea of the wide choice available for text recognition. Text interpretation may be difficult because abbreviations can vary in accordance with local habits, so this part may require specific training of the DL procedures, and a heavier interactive step.

Regarding line segment detection (to determine the relationships between relatives), many approaches are available in the literature. The critical points of line recognition are the problems of jaggedness, waviness, and similar unforeseen anomalies often present in hand-drawn graphs, which make recognizing line segments in hand-drawn diagrams a

challenging computer vision task. Several techniques and approaches can be employed to tackle this problem, such as various kinds of (possibly generalized) Hough Transforms, edge detection algorithms such as Canny or Sobel, RANSAC (Random Sample Consensus, a robust method for fitting models to data with outliers), and ML/DL models. A recent review paper [46], at the time of writing available as a preprint on the arXiv platform, is a good basis to start identifying effective procedures and available codes, because many of the reviewed papers also published their evaluation code. Another reference to extract inspiration and working code for line detection is https://www.catalyzex.com/s/Line%20 Segment%20Detection (accessed on 27 September 2023).

Regarding pedigree symbols, recognition can be achieved by DL approaches such as various R-CNNs (Region-based Convolutional Neural Networks) flavors or YOLO (You Only Look Once), or by ad-hoc methods based on image segmentation, feature calculation, and a Machine Learning classifier or some rule-based recognition algorithm.

The search for an optimal approach will start from the literature works focusing on the automatic recognition and digitization of graphs, flowcharts, piping, and instrumentation diagrams, etc., as they probably share many of the problems we will encounter for pedigree chart digitization.

The most critical preparatory step to the symbol recognition procedure will be software training, which of course needs a very rich and large annotated dataset of symbols. At present, to the best of our knowledge, no large publicly available database of annotated pedigree symbols exists, so this step necessarily involves the creation of our own dataset. We intend to proceed by following one or more of the following three different paths:

1. Annotating part of the pedigree charts from our clinical database, with existing annotation software (e.g., Roboflow Annotate, https://roboflow.com/annotate; Label Studio, https://github.com/HumanSignal/label-studio (accessed on 27 September 2023); Make Sense, https://www.makesense.ai/) (accessed on 27 September 2023);
2. Building from scratch a new dataset of hand-drawn symbols, containing the various necessary shapes for pedigree charts;
3. Taking advantage of existing datasets containing hand-drawn shapes (circles, squares, etc.) and deriving from each shape, by image processing techniques, a number of pedigree chart symbols according to the NSGC-PSWG standard, for deceased persons and for persons affected/non affected by the disease; for example, circles can be used as follows:

   (a) As they are (to denote alive women not affected by the disease);
   (b) After filling (for deceased women);
   (c) After superposing a diagonal line segment (for alive women affected by the disease).

Some free datasets available online have already been identified, such as https://github.com/frobertpixto/hand-drawn-shapes-dataset/ (accessed on 27 September 2023). The sizes of the datasets can then be increased by augmentation techniques (https://www.datacamp.com/tutorial/complete-guide-data-augmentation) (accessed on 27 September 2023), in particular, by small angle rotations, noise addition, distortion, etc.

## 2.4. Connectivity Recognition and Structure Analysis

This step consists of modeling the pedigree chart by the automatic reconstruction of the relationships between pedigree symbols, edges, and text parts (how are symbols connected by the lines? Where do lines intersect? To which symbol or line segment is each part of text logically connected?). The result will be an undirected graph encoding the family pedigree, where the nodes are the pedigree symbols or correspond to junctions between lines (see Figure 3 for the details), the edges represent family relationships, and both nodes and edges have convenient attributes storing the details of the relationships between family members. Proximity check and rule-based algorithms will be used to establish the relationships between symbols, edges, and text.

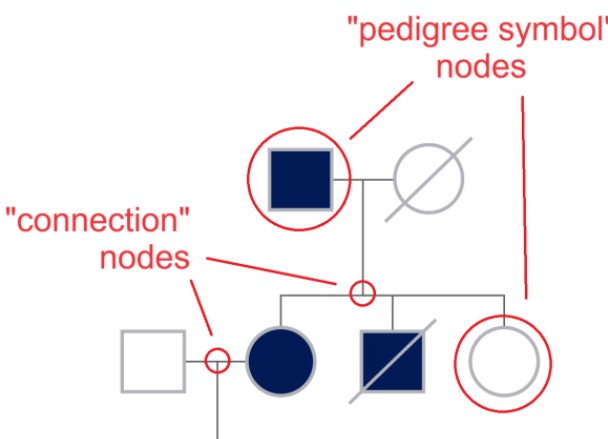

**Figure 3.** A graph G = (V, E) consists of a set of vertices or nodes (V) and a set of edges €. A vertex represents an endpoint of an edge, and an edge joins two vertices. In order to model the pedigree chart as an undirected graph, in addition to the nodes represented by the symbols, supplementary nodes can be introduced at the connection points between the line segments, so that V = {pedigree symbol nodes, connection nodes} and E = {suitably split line segments}. Nodes and edges will have convenient attributes storing the details of the relationships between family members. The figure clarifies the creation of new nodes, showing some of the nodes by red circles.

### 2.5. Saving to a Standard Format

Standard file formats exist for digital pedigree charts. Some of the most known formats are .ped (https://csg.sph.umich.edu/abecasis/Pedstats/tour/input.html, https://gatk.broadinstitute.org/hc/en-us/articles/360035531972-PED-Pedigree-format), and gedcom (https://lisalouisecooke.com/2017/03/11/gedcom-file (accessed on 27 September 2023), Genealogical Data Communication). Saving in standard formats allows to exchange genealogical data between different software programs for analysis.

### 2.6. System for De Novo Digitization of Pedigree Charts

The digitization of pedigree charts from paper to digital media constitutes the central focus of our paper. Our software tool will also include the direct de novo digitization of pedigree charts. Through an intuitively designed user interface, our software will facilitate the construction of digital pedigree charts using standard symbology. An exemplary instance of such software in the existing literature is f-tree (https://holonic-systems.com/f-tree/en/) (accessed on 27 September 2023) [6], an open-source product developed by the School of Medicine at Iwata University in Japan. Programs like f-tree, or comparable alternatives, may serve as benchmarks or control tools to enable the comparison and validation of the outputs generated by our software. In this way, our system offers a groundbreaking feature by allowing users to input pedigree charts directly into the software without the intermediary step of transitioning from a paper format. This innovation revolutionizes a traditionally manual system heavily reliant on handwritten pedigree charts. The user-friendly interface streamlines the process, enhancing efficiency and expanding the scope of pedigree chart digitization beyond conventional paper-based methods.

### 2.7. Predicting At-Risk Individuals on the Digital Pedigree Chart

During the pre-genomic phase and OGC counseling, the software will indeed act as a CAD, allowing the semi-automatic digitization of pedigree charts from paper media, and supporting their completion and refinement by a user-friendly graphical interface, thus, automating and simplifying the chart creation process as much as possible. Now, by having a large number of digitized pedigree charts combined with their outcome of the genetic mutation test present for each, it will also be possible to train new AI algorithms that "learn" from the examples provided to recognize who carried the mutation. Once

trained, the CAD can indeed take a new digital pedigree chart (never seen before) as input and predict the at-risk individual(s) directly on the chart, automatically. The idea is to highlight the at-risk individual(s) with a bounding-box and also provide a risk percentage for each highlighted subject.

To this end, classifiers based on both typical ML paradigms (decision trees, support vector machines, naive bayes) and DL (CNN) or Transfer Learning (TL) (using already trained convolutional neural networks) will be designed and implemented. Thus, multiple learning models will be used for the same data source, ensuring great diversity between them. The algorithms will be appropriately evaluated with dimensionality reduction techniques and information content preservation.

The performance of the intelligent risk prediction model on pedigree charts will be compared with that obtainable from predictive models already mentioned and used in the clinical field. These models, based on statistical predictions, assess a certain probability using personal and family medical history. Among these, we find the BCRAT (Breast Cancer Risk Assessment Tool), also known as the Gail model, a breast cancer risk assessment tool developed to estimate a woman's risk of developing invasive breast cancer over the next 5 years and up to the age of 90. This tool uses a combination of risk factors, including age, reproductive history, family history of breast cancer, and previous breast biopsies. BOADICEA is another risk prediction model, estimating individual risk for breast and ovarian cancer based on family history and specific genetic mutations (like BRCA1 and BRCA2). Unlike the Gail model, BOADICEA considers the detailed family history, including first, second, and third-degree relatives, and can also incorporate genetic test results. Other commonly used models are IBIS, which provides the breast cancer risk using the Tyrer–Cuzick index, in percentage (https://ibis-risk-calculator.magview.com/) (accessed on 27 September 2023), and BRCAPRO, which instead returns the risk of having BRCA1/2 mutations (https://projects.iq.harvard.edu/bayesmendel/brcapro) (accessed on 27 September 2023).

Although these tools can be used to be compared with the results that will come from our CAD, recent studies [47] have highlighted that such statistical models have limitations in their predictive capacity, especially when compared with more advanced methods like Machine Learning algorithms on the same data. AI may indeed show significant capability in improving the accuracy of classifying women with and without breast cancer.

Precisely for this reason, the goal of our CAD system is to surpass the limitations of traditional statistical models, offering specialists a powerful and integrated tool that can identify high-risk patients more efficiently, allowing for preventive interventions and personalized therapies. This integrated CAD will be a significant aid during OGC, providing the specialist with an initial indication of at-risk individuals along with an associated risk percentage. The specialized oncologist will then decide whether the individual should undergo mutational testing. Once the mutational status has been determined, individuals carrying the mutation(s) will enter the Surveillance Protocol for cancer diagnosis.

### 2.8. Preliminary Tests

Although this paper essentially reports a research plan and a project definition, with the purpose of sharing ideas and purposes and to stimulate collaborations, some preliminary tests were performed to obtain insights and suggestions about project feasibility. We started from the fundamental problem of symbol and text recognition followed by connectivity assessment, because once the basic graphic elements and text are detected and correctly connected, structural analysis can be faced by reconstructing attributed graphs, and the obtained trees can be used to build the necessary database for ML experiments.

The approach we started from was based on digital image processing and was developed in python with OpenCV, scikit-image, and EasyOCR.

Digital images of some test genealogical trees were obtained by scanning (image resolution around $2300 \times 1600$) and were initially preprocessed with a simple chain of filters (currently consisting of conversion to gray values, application of an adaptive threshold

filter, skeletonization) to pass from the raw images to their black and white versions (see Figure 4 for an example). Then, the images were fed to the recognition pipeline.

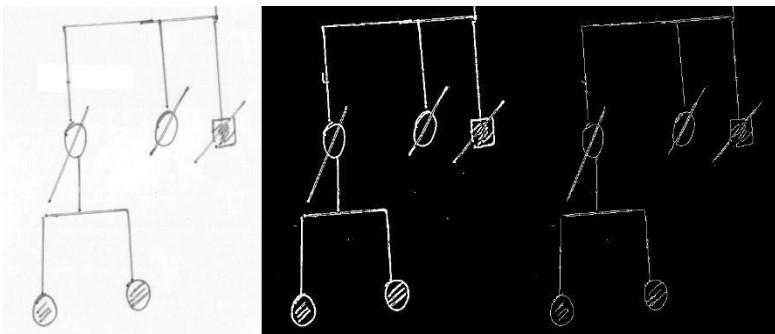

**Figure 4.** Example of preprocessing, from the original document scan (**left**), to the image obtained by an adaptive threshold filter (**center**), and finally to the skeletonized diagram (**right**). Some lines, in particular the oblique segments, appear as double after skeletonization because of the particular pen used, and give origin to double lines after the Hough transform. The defect is removed with line merging (see text and Figures 5 and 6).

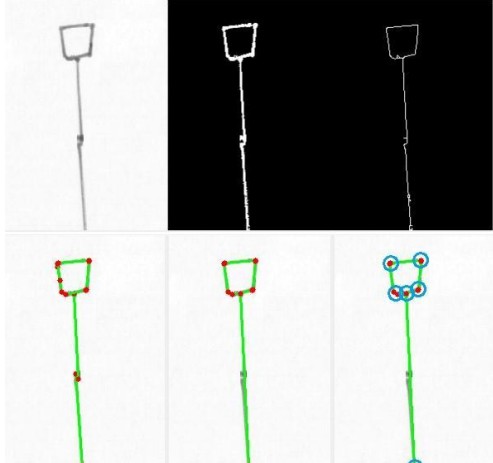

**Figure 5.** Example of preprocessing (**top row**) followed (**bottom row**) by the Hough transform for straight line segment detection (**left**; detected lines in green, endpoints in red), line merging (**center**; some lines were merged so that now only the essential five segments survive), and connection locating (**right**; found connections in blue).

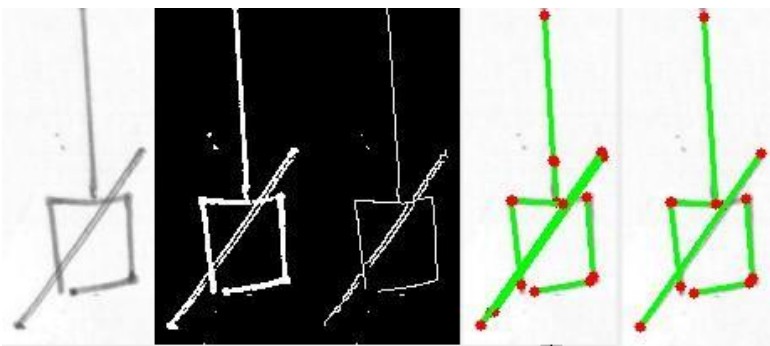

**Figure 6.** In this example, the problem of double lines due to the specific pen employed, is evident in the diagonal segment. The problem is solved by segment merging (rightmost picture). The diagram fragment was effectively reduced to the minimum number of segments (four for the square, one for the diagonal line, and one for the vertical line).

As the first step, a module for separating text from graphics (symbols and connection lines) was developed; in this way, an independent analysis of text and genealogical tree symbols could be performed and many problems of false positives in symbol detection due to text misinterpretation could be avoided.

Text windows were located by EasyOCR, which was also used to obtain a first guess of text meaning. It was soon clear that hand-drawn text recognition was a very tough task because text mainly consisted of some key words (which were often recognized by OCR, such as "melanoma"), first names (for which a database will be created and used for training or for a posteriori automatic correction), family names (more difficult to deduce), and specific abbreviations. The difficulty was expected and will be tackled in the interactive stage of the software, where a click on each text window will prompt the user to correct the text.

Then, we worked on line segments and, consequently, squares/rectangles. The Hough transform in OpenCV was used to find all the "straight" line segments. The algorithm was sensitive and found all the segments in the diagram, with much redundancy and with defects (fragmented or multiple segments). The problem of redundancy and the presence of defects were solved by implementing line merging [48] which was used to collapse similar segments into one, where similarity was defined in terms of tolerances in the angle and distance between the segments, both for overlapping and non-overlapping lines. After line merging, connections between the endpoints of different segments were found by simply locating the pairs of endpoints close to each other according to a defined threshold. On the other hand, connections between a segment endpoint and an internal point of another segment were also searched for, by assessing the closeness of the endpoint to each of the points of the segment, computed by Bresenham's line algorithm. The two types of connections were labeled in a different way, as the distinction is useful for structure analysis. See Figures 5 and 6 for two examples of line detection, line merging, and connection assessment.

The detection of squares/rectangles was tackled by searching for cyclic connections between quasi-horizontal (H) and quasi-vertical (V) segments (i.e., sequences like H-V-H-V in which connections between segment endpoints were previously discovered). The code was tested on a few sample images, demonstrating reasonable detection accuracy.

Finally, some tests were performed for circle/ellipse recognition. In this case, the number of false positives was quite high and a simple algorithm for ellipse merging (which was coded and tested) was not sufficient to reduce them and to sufficiently clean the result. In this case, a method of reduction of false positives, or a totally different approach, possibly based on ML, is in order; our tests are now going in this direction.

It is evident from what has been exposed that the methods which have been implemented only skim the surface of this very complex problem. It is also clear that many "magic numbers" (thresholds, tolerances) are at present only hard coded instead of being deduced from the image itself, making the system more adaptive must be one of next goals. Nonetheless, these feasibility tests suggest that recognition starting from simple image processing procedures (already explored in other works, such as [40]) is a good starting point, to be completed with more complex and smart approaches.

## 3. Results

The pedigree chart is the primary foundational tool in OGC. It is utilized to assess the risk an individual has of inheriting a mutation predisposing them to hereditary cancer. Thus, the pedigree chart is central to the entire research process. The processes of developing a CAD tool capable of digitizing paper-based pedigree charts or the automatic generation of new digitized ones followed by the prediction of high-risk individuals, as envisaged in our research plan, would introduce significant and competitive innovations in both process and service. The main outcomes will be defined in the next sections.

### 3.1. Digitizing Paper-Based Pedigree Charts

The first innovation of our CAD extends to the ability to digitize pedigree charts that have been previously collected in paper format. Traditionally, pedigree charts, valuable for understanding the family history of diseases, are managed in paper format, often entailing time-consuming and error-prone processes. Current hospital operational systems are based on a mix of generic software platforms related to the implementation of electronic health records, or other platforms—not necessarily open ones—present in the hospital. These systems are unable to represent diverse information in a structured manner (e.g., relationships among different individuals in a study), forcing medical personnel to manually collect data, including reconstructing pedigree charts on paper. This makes managing such complex data difficult, leading to information loss and rendering it unavailable for further analysis. In a context where healthcare service digitization is viewed as a pivotal turning point for the evolution of the sector, our software tool fully embraces this perspective and aims to simplify and enhance this process by building a software tool specifically designed for pedigree chart management. Beyond significant time savings, this transition reduces the risks associated with manual data entry errors, ensuring the accuracy and reliability of the collected information. Our tool will not only digitize existing paper charts but will also be capable of creating new ones from scratch. The introduction of this digitization represents a seamless transition for healthcare professionals, moving from traditional paper-based record keeping to an efficient digital data entry system.

### 3.2. Predicting High-Risk Individuals

Beyond digitization, our software tool will empower genetic oncologists to identify individuals within the pedigree chart who are at an elevated risk of genetic mutation predisposition. The software will also indicate the percentage of risk for each individual directly on the pedigree chart. To the best of our knowledge, there is no existing AI model in the literature that functions as a CAD system capable of predicting the percentage of this risk directly on the digital pedigree chart, whether newly digitized or derived from a paper-based source. The prediction of at-risk individuals could significantly support physicians during OGC in selecting ideal candidates for genetic testing. The advantage our CAD tool would offer should be considered keeping in mind that determining the appropriateness of a genetic test for suspected hereditary cancer is challenging, particularly for the hereditary forms of the most common tumors in the population. When there are no pathognomonic characteristics of hereditary disease but only data on the prevalence of the mutation in subgroups of cases selected for different criteria, the criteria for accessing genetic tests represent a compromise between containing costs and offering a test that can significantly impact prevention possibilities. These criteria are often superimposable on the criteria for sending individuals to OGC. A distinctive aspect of OGC is the need to involve other family members beyond the individual seeking consultation. This occurs in the preliminary phase of the approach to choose the most suitable family member for the search for a possible "unknown mutation". When the individual seeking OGC is a healthy person, it is typically proposed to involve a close relative who has already developed the disease. The result of the genetic test will then inform decisions about the presence or absence of predisposition in the family. Analyzing the family member with the highest likelihood of mutation first allows, if the genetic test is normal, the conclusion that the family's oncological history is probably not attributable to mutations in the analyzed gene. If the genetic test is positive, all willing family members can benefit from the test to identify the "specific mutation", thereby reducing the costs and time of the diagnostic process. Conversely, if a healthy family member is analyzed first and the genetic test for the search for an "unknown mutation" is normal, it becomes challenging to determine whether the subject has not inherited the mutation present in the family or if there is no mutation in the family (in the analyzed gene/s). The estimate of the oncological risk must consider this limitation of the test. In current practice, selecting the most suitable candidate for the family's diagnostic genetic test can be exceedingly difficult or even impossible,

sometimes due to reasons beyond the control of the person undergoing OGC (e.g., no living or willing patient to undergo the genetic test). In this context, the sinnovative approach of a software able to recognize high-risk individuals represents a pioneering initiative aimed at addressing and rectifying challenges and disparities in the current landscape of OGC and genetic testing; it will aid clinical geneticists and oncologists in selecting candidates, potentially allowing for an evidence-based approach. This is especially crucial given the inherent complexity and variability of genetic variants encountered in large populations.

### 3.3. Future Extensibility: The Potential of a Clinical Decision Support System

During the pre-genomic phase and OGC counseling, our CAD is thought to (1) perform the semi-automatic digitization of pedigree charts from paper media, and support their completion and refinement (but also the creation from scratch) by a user-friendly graphical interface, thus automating and simplifying the chart creation process; (2) predict subjects at risk of carrying the mutations and their risk percentage on these digital pedigree charts, while also considering national recommendations.

Our CAD system can also be envisioned as a comprehensive product that integrates the functionalities of a CAD with those of a Clinical Decision Support System (CDSS), serving as a predictive model for disease management. To elucidate this capability, a premise is useful, namely, in the post-genomic phase, upon obtaining the genetic test results, only individuals exhibiting the mutation—thus considered at risk—are enrolled in the Surveillance Protocol. Within this protocol, they are encouraged to undergo screenings for the diagnosis of hereditary tumors. Specifically, at-risk individuals undergo various imaging procedures, such as Magnetic Resonance Imaging (MRI) and mammography for breast cancer, and ovarian ultrasound for ovarian cancer. Individuals testing negative in the screenings are monitored over time, repeating periodic imaging exams. On the other hand, those testing positive in the screenings and diagnosed with cancer are overseen by oncologists, and they may undergo surgery or radiotherapy after medical imaging assessment and biopsy, which is considered the gold standard for diagnosis in this context. This represents the standard clinical procedure for at-risk patients.

Looking ahead, as a significant volume of medical images is generated, our planned software would have the potential to integrate additional AI algorithms. These algorithms can serve as predictive models, responding to specific questions for which they have been trained. Consequently, they can predict new prognostic variables even in the post-genomic test period, assisting oncologists in managing individuals within the Surveillance Protocol for cancer screening. During this phase, the software functions as a CDSS. These intelligent models can be trained to predict new prognostic variables or support genetic oncologists in managing cancer lesions. Some of these possibilities include the following:

-   Automatic lesion segmentation: Medical images could be analyzed by expert radiologists and manually outlined. These annotated images can be used to train an AI system to autonomously identify lesions in new images. Once optimized, this model could be integrated into the software, assisting oncologists in operator-independent lesion localization, contouring, and size assessment.
-   Radiomics and radiogenomics analysis: After identifying a lesion, radiomics analyses can be conducted, extracting quantitative information from images that is not immediately visible to the human eye. This "feature extraction", combined with genetic data, would provide a more in-depth view of a patient's specific tumor. This advanced model, once developed, could be integrated into the software, aiding in tumor diagnosis and characterization.
-   Therapeutic monitoring: In the post-diagnosis phase, if a tumor is detected, medical images can serve as a crucial tool to monitor treatment efficacy. These images can be used to train predictive systems aimed at tracking therapy progress over time. For example, by comparing automatically segmented lesions before and after treatment, quantitative differences can be assessed to determine if there has been a reduction in tu-

mor size. This data-driven approach allows for a more precise evaluation of treatment outcomes and helps clinicians make informed decisions regarding patient care.

- Risk candidate identification: Those with genetic mutations might not necessarily manifest a tumor during their lifetime. Ideally, by analyzing omics data collected from each individual, such as genomics, transcriptomics, proteomics, metabolomics, pathomics, and radiomics, and observing whether they develop the disease in specific time intervals (e.g., 2, 3, or 5 years), an advanced system could be trained to predict cancer risk. This system, using a combination of clinical, genetic, and other data, could determine the circumstances under which an individual might fall ill. Once optimized, the system could assess risk based on the data of an individual undergoing these tests for the first time. This would not only provide a tool for risk estimation but also identify the primary risk factors. Once developed, this model could be integrated into the software, providing evaluations based on an individual's omics data and risk factors.

These are just a few examples. Naturally, only with data analysis will it be possible to understand which other outcomes could be achieved. The models thus trained can be integrated into the software and function as a CDSS in the post-genetic test phase, providing enormous value in the Surveillance Protocol for patient management. This extension of our CAD, connects the fields of imaging and genetics, providing a holistic and personalized approach to patient care.

## 4. Discussion

While it is widely recognized that 5 to 10% of breast cancers and up to 15% of ovarian cancers exhibit a hereditary predisposition [49], the pressing challenge lies in uncovering these cases. There is an urgent need to pinpoint individuals at risk who should undergo genetic testing, along with their at-risk family members, to determine their susceptibility to hereditary–familial tumors.

In the era of omics sciences and big data, the application of AI principles in medicine, especially in oncology, is already a reality. ML and DL techniques are used today to predict therapy responses, disease progression, early radiological identification of the disease, and the discovery and validation of new diagnostic biomarkers. Nevertheless, some types of health data have remained on the sidelines of this remarkable revolution. An example is the pedigree charts in genetic oncology, crucial for identifying individuals within a family—based on current national and international guidelines—who can be evaluated to assess their genetic predisposition to an increased risk of developing new neoplastic diseases.

As far as we know, there are no examples in the literature today about the use of AI-based technologies on these datasets, which essentially remain in the patient's and their family's paper or electronic file, without further use. The development of an automated CAD system for the digitization of hand-drawn pedigree charts and the prediction of at-risk individuals, is a complex and promising endeavor poised to reshape the landscape of OGC, clinical research, and patient care within the realm of genetic oncologists.

Our CAD should be seen in the broader context of the European Union. The European Union views the digitization process as an essential tool in the service of cancer care (The European Digital Strategy—Shaping Europe's digital future (europa.eu)). The development of strategies to overcome the "barriers" opposing a full digital transformation and the exploitation of the resulting data in terms of interoperability is deemed indispensable for clinical care, research, and planning purposes. Digital health encompasses all Information and Communication Technologies (ICTs) required for the functioning of the health system (from electronic prescriptions to telemedicine and telecare, to information supporting epidemiological studies and clinical research). In Italy, the transition to digital health is one of the prerequisites for achieving the country's health objectives, helping to simplify access to health and social care services and redesigning a national health service model that guides the patient in using health services, meeting their needs while containing

costs. This need is acknowledged by the State-Regions Conference and incorporated into the verification of Essential Levels of Care. To this end, it is essential to ensure national governance of the digitization process with a strategic, systemic, and integrated vision that, thanks to coordinated and flexible technical protocols, allows the interoperability of ICT systems, reducing the risk of local misalignments. The transition to digital health, acting transversally across various areas, supports different organizations in accelerating the achievement of strategic objectives, namely reducing the incidence of tumors, improving diagnosis and treatment, reducing cancer mortality, and enhancing the quality of life of patients and long-term survivors. Digitizing healthcare services is then a crucial step towards the modernization of the healthcare system.

Turning now to the specifics of the Italian context, it is also important to mention the "National Oncological Plan" [50] approved on 27 January 2023, with the State-Regions agreement. It is a planning and guidance document for the prevention and fight against cancer from 2023 to 2027. This Plan identifies objectives and strategic lines consistent with the European Plan against Cancer (Europe's Beating Cancer Plan) [51] and must be embraced by the Regions and Autonomous Provinces with their own measures, adopting the most suitable organizational solutions in relation to the needs of their own programming. This document aims to define a global and cross-sectoral approach to reduce human suffering and the socio-economic burden of tumors. According to the plan, screening and personalized care for subjects at high hereditary–familial risk should be emphasized. This plan, in fact, considers it necessary to pursue the personalization of preventive actions through the identification of high-risk subjects and the establishment of intensified surveillance and specific prevention programs that complement screenings, integrating with them from a structural and operational point of view. It is clear that there is a need to intervene with strategies aimed at increasing the number and quality of life of people at risk. In addition, the document insists on the role of prevention based on the identification of disease determinants, on quantifying risk, and on recognizing the role of genetic and environmental components as factors contributing to the onset of the disease.

In this scenario, early detection of hereditary breast and ovarian cancers targeting the BRCA1/2 gene mutation is of absolute importance, especially since drug therapy with PARP inhibitors is already available for these mutations. However, although the National Plan from the Italian Minister of Health for 2014–2019 and the subsequent one for 2020–2025 [52] anticipated the adoption of organized pathways for the prevention of breast and ovarian cancer associated with pathogenic variants of the BRCA genes in all Italian Local Health Authorities by 2019, in reality, the Clinical Pathway "High Hereditary-Familial Risk for carriers of pathogenic BRCA variants" has not yet been approved in all regions. Only in some is the exemption from the ticket payment recognized for surveillance tests of high-risk healthy individuals, as reported in the latest National Plan for 2023–2027 [50]. Similar evaluations can obviously also be carried out on other and less well-known CPGs, the list of which increases together with the development of new studies and innovative instrumental methods.

Recognizing the crucial role of organized pathways for the prevention and early diagnosis of hereditary cancers, a CAD capable of digitizing pedigree charts and predicting at-risk subjects, aligns perfectly with the objectives set by the National Plans. Additionally, in the future, the tool can function as an integrated system to monitor mutation carriers, serving as CDSS in the post-genomic test phase, assisting the genetic oncologist in managing the individual within the Surveillance Protocol to assess the presence of the tumor and its prognosis. Not only will this expedite the risk identification process functioning as a CAD, but by integrating medical images and omics data, it will provide a comprehensive and personalized risk profile of that at-risk individual.

We aim to bridge the gap between scientific advancements and practical implementation by modernizing and enhancing existing OGC services currently in place across the country. The integration of our innovative software into existing healthcare systems represents a significant step towards achieving the goals outlined in the national plans and

also a proactive approach to improving the overall health and well-being of individuals at high risk.

## 5. Conclusions

Considering the tens of millions of cases diagnosed every year throughout the world [1] and that in approximately 10% [3] (and perhaps even more for some types of cancer) of these a hereditary mechanism is involved, we can understand how the problem is not negligible in numerical terms and how important it is to introduce innovations in the clinical practice and management of this fraction of cancer patients.

Traditionally, pedigree charts, essential for identifying individuals with an increased risk of developing hereditary tumors, have been maintained in physical form, often resulting in time-consuming and error-prone processes. Our plan is to introduce and automate a CAD system that can digitize existing hand-drawn pedigree charts. This involves the recognition of symbols, line segments, text parts, the reconstruction of the pedigree chart structure, and their de novo creation, heralding a significant process and service innovation in the medical care workflow. The introduction of this digitization process presents healthcare professionals with a seamless transition from the conventional paper-based record keeping to efficient and error-reducing digital data entry. Beyond the significant time savings, this transition mitigates the risks associated with manual data entry errors, ensuring the accuracy and reliability of the collected information.

In addition, our CAD tool would hopefully predict a genetic predisposition risk directly from these digital pedigree charts. This new system has the potential to streamline and enhance various aspects of genetic counseling, research, and healthcare, leading to improved patient care and advancements in the field of oncogenetics. By incorporating medical images and other outcomes from omics sciences, there is also a fertile ground for training additional AI systems, broadening the software predictive capabilities. In conclusion, by focusing on the digitization of paper pedigree charts and the prediction of high-risk individuals, we aim to revolutionize the way familial and hereditary cancer risk information is collected and managed.

**Author Contributions:** Conceptualization, L.C., E.R. and G.D.N.; methodology, L.C., E.R. and G.D.N.; validation, T.G., F.B., E.D.M. and G.D.N.; formal analysis, L.C.; investigation, L.C. and E.R.; resources, G.D.N.; data curation, L.C., E.R., T.G., F.B., E.D.M. and G.D.N.; writing—original draft preparation, L.C., E.R. and G.D.N.; writing—review and editing, T.G., F.B., E.D.M. and G.D.N.; supervision, T.G., F.B., E.D.M. and G.D.N. All authors have read and agreed to the published version of the manuscript.

**Funding:** This research received no external funding.

**Institutional Review Board Statement:** Not applicable.

**Conflicts of Interest:** The authors declare no conflicts of interest.

## List of Abbreviations

| Abbreviation | Meaning |
| --- | --- |
| BCRAT | Breast Cancer Risk Assessment Tool |
| BOADICEA | Breast and Ovarian Analysis of Disease Incidence and Carrier Estimation Algorithm |
| CAD | Computer-Aided Detection/Diagnosis |
| CASE | Computer-Aided Software Engineering |
| CDSS | Clinical Decision Support System |
| CLEF-IP | Conference and Labs of the Evaluation Forum—Intellectual Property |
| CNN | Convolutional Neural Networks |
| CPG | Cancer Predisposition Gene |
| CT | Computed Tomography |
| DL | Deep Learning |

| | |
|---|---|
| DNN | Deep Neural Network |
| FR-DETR | Flowchart Recognition Detection Transformer |
| GUI | Graphical User Interface |
| GWAS | Genome-Wide Association Studies |
| HBOC | Hereditary Breast and Ovarian Cancer |
| IBIS | International Breast Cancer Intervention Study |
| ML | Machine Learning |
| MRI | Magnetic Resonance Imaging |
| NGS | Next Generation Sequencing |
| NSGC | National Society of Genetic Counselors |
| OCR | Optical Character Recognition |
| OGC | Oncological Genetic Counseling |
| P&ID | Piping and Instrumentation Diagram |
| PSTF | Pedigree Standardization Task Force |
| PSWG | Pedigree Standardization Work Group |
| RANSAC | Random Sample Consensus |
| R-CNN | Region-based Convolutional Neural Network |
| TL | Transfer Learning |
| UML | Unified Modeling Language |
| YOLO | You Only Look Once |
| YOLSO | You Only Look for a Symbol Once |

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
