# Peer review of "Artificial Intelligence Techniques and Pedigree Charts in Oncogenetics: Towards an Experimental Multioutput Software System for Digitization and Risk Prediction"

_computation, doi:10.3390/computation12030047_

Round 1

Reviewer 1 Report

Comments and Suggestions for Authors

Title of the manuscript: "Artificial intelligence techniques and pedigree charts in oncogenetics: towards an experimental Computer Assisted Detection/Diagnosis (CAD) multioutput system for digitization and risk provisions"

The authors propose a system for Computer Assisted Diagnosis that uses Machine Learning and Deep Learning techniques. This system is designed to help genetic oncologists in converting paper-based pedigree charts into digital format, generating new digital charts, and automatically predicting the risk of genetic predisposition directly from these digital charts.

I have the following comments:

The authors have extensive expertise in both the medical and technical aspects of the proposed system.

This article is intended as a research plan. It is justified to publish the plan at this stage so that it is possible to give comments and others can coordinate their research in a compatible way.

Minor issues:

1) There are some layout issues that the authors should check.

a) The use of abbreviations in the abstract.

b) The use of "Background:", "Methods:", "Results:", and "Conclusion:". Please check the instructions for authors (https://www.mdpi.com/journal/computation/instructions).

c) Please check: line 247 "Text.", line 262 "Line segments.", line 303 "and so on."

d) Please check lines 335 and 337 "de-novo", "de novo".

e) There are very long paragraphs, for example, lines 117-174. The paragraph could be divided into two or more paragraphs.

and so on.

2) The title of the paper is quite long. You can consider shortening it.

3) The introductory section is quite technical. Consider moving content to new sections or other sections.

4) It is possible to add a list of abbreviations at the end of the paper.

Author Response

Thank you for your helpful comments.

Minor issues:

1) There are some layout issues that the authors should check:

  1. a) The use of abbreviations in the abstract.

We eliminated the abbreviations (acronyms).

  1. b) The use of "Background:", "Methods:", "Results:", and "Conclusion:". Please check the instructions for authors (https://www.mdpi.com/journal/computation/instructions).

We eliminated the titles of the various sections.

  1. c) Please check: line 247 "Text.", line 262 "Line segments.", line 303 "and so on."

We eliminated the sections.

  1. d) Please check lines 335 and 337 "de-novo", "de novo".

Corrections made.

      1. e) There are very long paragraphs, for example, lines 117-174. The paragraph could be divided into two or more paragraphs.

We divided the original paragraph into several ones in order to improve reading and comprehension.

2) The title of the paper is quite long. You can consider shortening it.

We shortened and modified the title, leaving its message unchanged.

3) The introductory section is quite technical. Consider moving content to new sections or other sections.

We know that the part of the introduction where we discuss the limits relating to symbol recognition is rather technical, but we thought we would leave it as it is because it probably wouldn't have been appropriate in other sections.

4) It is possible to add a list of abbreviations at the end of the paper.

We added a list of abbreviations at the end of the manuscript.

Reviewer 2 Report

Comments and Suggestions for Authors

This report lacks essential foundational data and initial analyses to substantiate the authors' assertions. Prior to developing the system, it is imperative that critical analyses and pretests are conducted to demonstrate the feasibility of the proposed elements.

The method sections lack basic information on statistical results or empirical evidence supporting the automatic pedigree chart reconstruction, as well as the automatic recognition of symbols, edges, and text.

Furthermore, the report fails to present any data or preliminary analysis results showcasing the training and validation processes of the machine learning models. Significantly, there is a notable absence of information on model evaluation and comparison, leaving reviewers without evidence to assess the feasibility and validity of the proposed system.

Comments on the Quality of English Language

This report needs moderate revision to enhance the language.

Author Response

Thank you for your valuable feedback and the suggestion to include tests on the machine learning component of our project. At present, our database is still in the preparation phase, with considerable manual effort being dedicated to reconstructing decision trees by hand. Despite these challenges, we have initiated tests with classical image processing techniques, the results of which are now documented in the paper. Additionally, we are exploring deep learning methods. However, these efforts remain in the preliminary stages, as the development of a comprehensive dataset of symbols, crucial for model training, is ongoing.

Our project is primarily presented as a research plan intended to solicit collaborations and stimulate ideas within the field. Hence, we believe that an excessive focus on the ML component—at this juncture—might detract from the core essence of our article. We aim to show that a feasibility study is underway, yielding promising results and highlighting challenges as we progress.

Reviewer 3 Report

Comments and Suggestions for Authors

Review this sentence: Specifically, at-risk individuals undergo various imaging procedures, such as Magnetic Resonance Imaging (MRIs) and mammography for breast cancer, or ovarian ultrasound for ovarian cancer. please use and not or ovarian ultrasound and mammogram are different (you can check very first sentence of discussion section). 

This paper begins by presenting the facts regarding cancer cases, totaling around 18.1 million new cases, in the introduction sections up to the 3rd paragraph, and the 4th paragraph focuses on AI techniques. The authors then delve into explaining a major limiting factor in paragraph 5 (Page 2) concerning symbol recognition in hand-drawn graphs, extending up to page number 9 where a breast cancer risk assessment tool is developed. They describe various elements such as UML, CASE (Computer-Aided Software Engineering), shape, size, and hand-drawn pedigrees. To enhance the discussion, it is recommended to identify influential papers demonstrating how CAD systems aid in facilitating medical care flow and to present the findings in a table.

Upon reviewing the conclusion section, it is observed that cancer is somewhat disconnected and described in a commonplace manner. Connecting the conclusion more seamlessly with the central ideas is advisable. Additionally, incorporating quantitative data to substantiate the presented ideas would bolster the paper's overall acceptability.

Lastly, it is advisable to compile all content resource links in the reference section for clarity and proper citation.

Author Response

Thank you for your helpful comments. 

Review this sentence: Specifically, at-risk individuals undergo various imaging procedures, such as Magnetic Resonance Imaging (MRIs) and mammography for breast cancer, or ovarian ultrasound for ovarian cancer. please use and not or ovarian ultrasound and mammogram are different (you can check very first sentence of discussion section).

Correction made.

This paper begins by presenting the facts regarding cancer cases, totaling around 18.1 million new cases, in the introduction sections up to the 3rd paragraph, and the 4th paragraph focuses on AI techniques. The authors then delve into explaining a major limiting factor in paragraph 5 (Page 2) concerning symbol recognition in hand-drawn graphs, extending up to page number 9 where a breast cancer risk assessment tool is developed. They describe various elements such as UML, CASE (Computer-Aided Software Engineering), shape, size, and hand-drawn pedigrees. To enhance the discussion, it is recommended to identify influential papers demonstrating how CAD systems aid in facilitating medical care flow and to present the findings in a table.

We added a new paragraph in the introduction to better include CAD systems along with a table where are reported the most influential papers about CAD tools in oncology published last year.

Upon reviewing the conclusion section, it is observed that cancer is somewhat disconnected and described in a commonplace manner. Connecting the conclusion more seamlessly with the central ideas is advisable. Additionally, incorporating quantitative data to substantiate the presented ideas would bolster the paper's overall acceptability.

We added an introductory part in the conclusions to better connect it to the central idea.

Lastly, it is advisable to compile all content resource links in the reference section for clarity and proper citation.

We modified all the references, adding the missing DOIs and URLs and adapting them to the MDPI style.

Round 2

Reviewer 2 Report

Comments and Suggestions for Authors

I appreciate the authors' responses and revisions. The incorporation of preliminary tests offers more informative insights into the feasibility of the proposed project. Nevertheless, I urge the authors to contemplate incorporating additional details regarding the statistical outcomes of subsequent experiments or assessments of model training and performance in the pretests. Such endeavors will significantly bolster the integrity and validity of this report. I don't need to see the revision again if the authors determine to make edits on this.

Comments on the Quality of English Language

Minor edit of the language is recommended.

Author Response

We have revised the English, in particular in the abstract, introduction and conclusions: some repetitions have been eliminated and several terms have been added or replaced to increase the understanding and fluency of the text. As regards the preliminary results, we did not consider studying them in depth from a statistical point of view at the moment because they are completely provisional and rapidly evolving as we test new techniques. Anyway, we thank the reviewer for his proposal and suggestion.